# Prolonged Water-Only Fasting Followed by a Whole-Plant-Food Diet Is a Potential Long-Term Management Strategy for Hypertension and Obesity

**DOI:** 10.3390/nu16223959

**Published:** 2024-11-20

**Authors:** Evelyn Zeiler, Sahmla Gabriel, Mackson Ncube, Natasha Thompson, Daniel Newmire, Eugene L. Scharf, Alan C. Goldhamer, Toshia R. Myers

**Affiliations:** 1TrueNorth Health Foundation, Santa Rosa, CA 95404, USA; drevezeiler@gmail.com (E.Z.); macksonncube@gmail.com (M.N.);; 2School of Health Promotion and Kinesiology, Texas Women’s University, Denton, TX 76204, USA; 3Department of Neurology, Mayo Clinic, Rochester, MN 55905, USA; 4TrueNorth Health Center, Santa Rosa, CA 95404, USA; dracg@truenorthhealth.com

**Keywords:** hypertension, prolonged water-only fasting, prolonged fasting, adverse events, whole-plant-food diet, anti-hypertensive medication use, obesity

## Abstract

**Objective:** This single-arm, pre–post interventional trial (clinicaltrials.gov, NCT04515095) investigates the safety, feasibility, and potential effectiveness of prolonged water-only fasting followed by a whole-plant-food diet in the long-term management of hypertension and other cardiometabolic disorders. **Methods:** Safety was assessed based on adverse events (AEs) that were recorded according to Common Terminology Criteria for Adverse Events (CTCAE) v5.0. Feasibility was assessed based on retention rate, ability to complete minimal fast length, and intervention acceptability. Twenty-nine participants with stage 1 and 2 hypertension and without type 2 diabetes were enrolled from a residential fasting center. **Results:** Study retention was 100% at the end of the refeed and 93% at the six-week follow-up visit. Median (range) fasting and refeeding duration were 11 (7–40) and 5 (3–17) days, respectively, and 90% of participants were able to complete at least 7 days of fasting. The majority of AEs were mild (grade 1) and transient and there were no higher-grade or serious AEs. At the end of the intervention, median systolic/diastolic blood pressure had normalized to below 130/80 mmHg, body weight reduced by >5%, and anti-hypertensive medication was completely discontinued. These results were sustained for at least six weeks and potentially up to one year. **Conclusions:** Our data suggest that the intervention may be a feasible, well-tolerated, low-risk option for lowering and managing high blood pressure, excess body weight, and other cardiometabolic disorders in people with stage 1 and 2 hypertension.

## 1. Introduction

More than half of US adults have hypertension (HTN), a primary modifiable risk factor for developing cardiovascular disease (CVD), kidney disease, and other conditions, with associated healthcare costs estimated at USD 131 billion annually [1,2,3,4,5]. Current treatment guidelines recommend diet and lifestyle modifications for all people with high blood pressure (BP) and the use of anti-hypertensive medications when systolic blood pressure (SBP) and/or diastolic blood pressure (DBP) reaches ≥140/90 mmHg (Stage 2 HTN) or in high-risk patients when SBP and/or DBP reaches ≥130/80 mmHg (Stage 1 HTN) [5]. However, up to 60% of medicated patients with HTN continue to have uncontrolled high BP, potentially due to modest BP reductions, high rates of adverse side effects, and low adherence rates associated with anti-hypertensive medications [6,7]. Additionally, lowering high BP with anti-hypertensive medications significantly reduces risk only in patients already at high risk for CVD [8]. Diet and lifestyle modifications may lower high BP, reduce associated CVD risks, and decrease rates of polypharmacy, but these interventions also suffer from low adherence rates [9,10,11]. Thus, there is a need for innovative treatment options to prevent and reverse high blood pressure, ideally with an overall improvement in cardiovascular health.

Medically supervised, prolonged water-only fasting followed by an exclusively whole-plant-food diet free of added salt, oil, and sugar (SOS-free diet) has a very low risk of causing severe or serious adverse events and effectively lowers borderline and high blood pressure [12,13,14]. The intervention may support adherence to diet and lifestyle modifications and also correlates with sustained improvements in biomarkers of CVD risk [15,16]. To further evaluate the potential of this intervention as a treatment for HTN, we conducted a single-arm pre–post interventional trial with long-term follow-up. The primary aim was to assess the safety and feasibility of the intervention in adults with stage 1 and stage 2 HTN and without type 2 diabetes. We systematically collected and reported adverse events and treatment deviations, along with study retention rates, participants’ ability to complete the minimum fasting duration, and self-reported acceptability of the intervention. Additionally, we measured the immediate, sustained, and long-term effects of the intervention on blood pressure and other cardiometabolic risk factors.

## 2. Materials and Methods

### 2.1. Ethical Approval

This study was approved by the TrueNorth Health Foundation Institutional Review Board (TNHF2020-1HTN and TNHF2021-1HTNFU) and registered at clinicaltrials.gov (NCT04515095). The research was conducted in accordance with the approved protocol and complied with the standards of the Declaration of Helsinki. All participants provided written informed consent before data collection began.

### 2.2. Participant Enrollment and Study Protocol

This study was a single-arm pre–post intervention trial that recruited patients with stage 1 and 2 HTN who were undergoing an elective, medically supervised, water-only fast of at least seven days at a residential fasting center between August 2020 and September 2021. Thirty participants were enrolled, of which one was ineligible due to pre-existing hyponatremia that contraindicates fasting (Figure 1). Consenting participants of any sex, aged 30–75 years, with uncontrolled (SBP ≥ 130 mmHg and/or DBP ≥ 80 mmHg) or medication-controlled HTN, fasting glucose < 126 mg/dL and/or hemoglobin A1c < 7%, and prior approval by a non-research physician to water-only fast for at least seven consecutive days followed by a refeeding period of at least half the fasting length were eligible for inclusion. Exclusion criteria included SBP > 180 and/or DBP > 120 mmHg at the time of enrollment, active malignancy, active kidney disease, active inflammatory disorder (including classic autoimmune connective tissue disorders, multiple sclerosis, inflammatory bowel disorders), stroke or heart attack within the last 90 days, and the inability to discontinue medications and/or supplements. Study dates overlapped with NCT04514146, conducted at the same fasting center, and 18/29 participants were co-enrolled in that study [15].

During the treatment period, onsite study visits occurred daily and at baseline (BL), end-of-fast (EOF), and end-of-refeed (EOR). Two additional study visits occurred either onsite or remotely: an expected visit that occurred 6 weeks after departure from the fasting center (6wFU) and one unexpected visit that occurred 12 months after the 6wFU visit (12mFU). During daily visits, symptoms and vital signs were assessed, and adverse events (AEs) were recorded. Additionally, demographics, medical diagnoses, medication use, total body weight (BW), blood pressure (BP), abdominal circumference (AC), and blood and urine samples were collected at BL, EOF, EOR, weekly from BL to EOR, 6wFU, and 12mFU visits (see supplemental methods 1.1 and 1.2 for data collection details). Web-based versions of three questionnaires were administered: the SOS-Free Diet Screener [15] at BL, 6wFU, and 12mFU visits; the Treatment Adherence/Acceptability Scale (TAAS) [17] questionnaire at EOF and 6wFU visits; and the Food Acceptability Questionnaire (FAQ) [18] at BL, EOR, 6wFU, and 12mFU visits. Data collection and survey distribution were performed using the web-based software Research Electronic Data Capture (REDCap) [19]. Missing data are described in Figure 1 and the legend of respective tables.

### 2.3. Medically Supervised Water-Only Fasting Protocol

The prefeeding, fasting, and refeeding protocol took place at a residential fasting center. The protocol was previously described in detail [14]. Participants were approved to water-only fast by medical doctors not affiliated with this study after a thorough examination, which included detailed patient history, comprehensive physical exam, basic neurological and psychological status, complete blood count (CBC), comprehensive metabolic panel (CMP), urinalysis, and additional tests as clinically indicated, in order to rule out contraindications to fasting. Medication use while water-only fasting is contraindicated, and ability to safely discontinue medication use was a requirement for enrollment in this study. Participants received 24 h medical supervision during the entire treatment (i.e., prefeeding, water-only fasting, and refeeding). Vital signs were examined by medical personnel twice per day, and labs were ordered once per week or as requested by the attending physician.

#### 2.3.1. Prefeeding

A prefeeding period began at least two days prior to the water-only fast, during which participants eliminated all recreational drugs (e.g., coffee, alcohol, nicotine, etc.) and specific foods (i.e., grains, legumes, dairy, meat, seafood, added sugar, oils, and salt, and all processed foods) and ate only the provided raw/steamed fruits and vegetables. During this time, participants were weaned off anti-hypertensive and other medications as determined appropriate by the attending physician based on clinical protocol and according to the individual participant’s medical needs.

#### 2.3.2. Fasting

Participants remained onsite during the entire fasting period, were encouraged to engage in a small amount of low-intensity physical activity (i.e., slow walking and light stretching), and did not operate heavy machinery (e.g., automobiles). During the water-only fast, participants drank a minimum of 1.2 L/day of steam-distilled water. If medically indicated (e.g., due to blood electrolyte imbalance, hypoglycemia, or gastroesophageal reflux disease) or desired by the participant, vegetable broth (VB; 80 kcal per day) was consumed. VB has such an insignificant number of calories that it does not prevent ketosis or reverse ketosis once it is initiated and is assumed to be equivalent to water. The fast continued for the predetermined amount of time or was modified according to the individual patient’s needs. In cases of more severe adverse events or discomfort, the fast was temporarily or indefinitely broken with fruit and vegetable juice (500–600 kcal per day) or potato/zucchini blend (900 kcal per day).

#### 2.3.3. Refeeding

The refeeding diet was administered in five phases: (1) fruit and vegetable juice; (2) raw fruits/vegetables; (3) raw/steamed vegetables and fruits; (4) raw/steamed fruits and vegetables, grains, and nuts or avocado; (5) unrestricted whole-plant foods free of added salt, oil, and sugar. Each of the five refeeding phases lasted 1 day for every 7–10 days of water-only fasting, for a period of time lasting at least one-half of the total fast length (e.g., 10 days of fasting would correspond to 5 days of refeeding with 1 day on each of the five phases). If medically indicated (e.g., due to food intolerance), the refeeding protocol was modified [14]. Before leaving the center, participants received basic nutrition education and were instructed to continue the exclusively whole-plant-food, SOS-free diet for at least the next six weeks. There were no further instructions after the 6wFU visit.

### 2.4. Adverse Events Recording and Analysis

While prefeeding, fasting, and refeeding, AEs were identified in daily interviews conducted by trained clinical research personnel, daily physical (i.e., daily vital signs) examinations, weekly blood analysis (i.e., CBC and CMP), and weekly urinalysis. AEs were recorded by date and according to Common Terminology Criteria for Adverse Events (CTCAE) [20] based on symptom categorization and severity (grade 1, mild; grade 2, moderate; grade 3, severe; grade 4, life-threatening; grade 5, death). The CTCAE describes AE terms according to the Medical Dictionary for Regulatory Activities (MedDRA) as lowest-level terms (LLT) that are grouped into a system organ class (SOC). For terms not specifically listed in the CTCAE and listed under the MedDRA SOC as ‘other’, new terms were added and graded according to the CTCAE guideline that described the AE as best as possible. All AEs were reported regardless of attribution. Pre-existing conditions were considered an AE if reoccurring or increasing in severity at any time during the intervention. The same AE may have occurred multiple times for the same participant and was counted as a new event if it decreased or increased in grade or had previously resolved but subsequently returned. AE outcomes (i.e., resolved, persisted, unknown) were determined as follows: the outcome was considered “resolved” if the symptom cleared completely or the severity of the event decreased (e.g., G2 decreased to G1). The outcome was considered “persisted” if the AE did not resolve at EOR or 6wFU visit. The outcome was considered “unknown” if the interview or specific analysis was not repeated (only AEs determined by CMP and CBC were monitored at 6wFU) or if the participant failed to report back at the 6wFU visit. The total number of AEs for all participants during the entire intervention was counted and presented based on SOC, LLT, grade, and outcome. Additionally, the top 10% of total AEs experienced by participants were presented based on LLT, grade, and treatment phase (i.e., prefeeding, fasting, and refeeding). AEs were further categorized as the highest-grade adverse event (HGAE) during the entirety of each participant’s intervention.

### 2.5. Daily Vital Signs

Daily vital signs, including body weight (BW), BP, peripheral oxygen saturation (SpO_2_), and body temperature (BT), were collected onsite by trained clinical research staff every morning from BL through EOR. Baseline values were calculated from day 2 daily vitals data. Participants were asked to rest in a seated position with their arms elevated to heart level on a table or pillow for five minutes prior to having their BP (Welch Allyn-Connex ProBP 3400, Hill-Rom Holding Inc., Chicago, IL, USA), BT (Welch Allyn Sure Temp plus 690, Hill-Rom Holding Inc., Chicago, IL, USA), peripheral oxygen saturation (SpO_2_, Zacurate 500DL, Zacurate, Stafford, TX, USA), and pulse rate and rhythm (manually for 15 s or 1 min) measured. BW was self-reported using a floor scale (WW26, Conair LLC, Stamford, CT, USA). Participants were also observed and asked about hours of sleep, energy level, urination characteristics (dysuria, difficulty, changes), nausea, presyncope (i.e., near fainting), quantity of water intake, bowel movements (frequency, characteristics), and any other symptoms for the previous 24 h as well as about any unresolved complaints.

### 2.6. Questionnaires

#### 2.6.1. SOS-Free Dietary Screener

Adherence to the recommended SOS-Free Diet was assessed with a 27-question screener as previously described [15]. The SOS-Free Dietary Screener was administered online at BL, 6wFU, and 12mFU visits. A non-adherence score was calculated based on the consumption of inclusionary and exclusionary foods over the previous 30 days. The minimum non-adherence score is zero (i.e., 100% adherent), and the maximum non-adherence score is 82.5 (i.e., 0% adherent).

#### 2.6.2. Treatment Acceptability Adherence Scale

Acceptability of the treatment was assessed using the Treatment Acceptability/Adherence Scale (TAAS) [17]. The TAAS was administered online at EOF and 6wFU visits. The TAAS contains 10 questions with response options ranging from 1 (disagree strongly) to 4 (neither agree nor disagree) to 7 (agree strongly). Negatively worded questions (#3–5, #7–8, #10) were reverse scored, and total scores were obtained by summing all answers. Scoring ranged from 70 to 10, with higher scores indicating greater treatment acceptability. Individual questions are shown in Appendix A.

#### 2.6.3. Food Acceptability Questionnaire

Acceptability of foods permitted on the SOS-Free Diet was assessed using the Food Acceptability Questionnaire [18]. The FAQ was administered online at BL, EOR, 6wFU, and 12mFU visits. The questionnaire contains 10 questions with response options ranging from 1 (not at all) to 7 (extremely). Negatively worded questions (#4 and #8) were reverse scored, and total scores were obtained by summing all answers. Scoring ranged from 70 to 10, with higher scores indicating greater acceptability of permissible foods. Individual questions are shown in Appendix A.

### 2.7. Statistical Analysis

Data cleaning and statistical analysis were conducted using R, with descriptive statistics reported as medians and interquartile ranges (IQR) [21]. The statistical analysis aimed to estimate within-group changes between visits (e.g., BL, EOF, EOR, 6wFU, and 12mFU). The analysis employed random intercept generalized linear mixed-effects models, with clinical parameters as dependent variables and participant ID as the grouping variable. Age and sex served as fixed effect control variables in all models; baseline HTN status was also included for binary outcome models.

Continuous outcomes were analyzed using lmerTest version 3.1.3 [22], while binary and count outcomes used lme4 version 1.1.32 [23]. Binary data were modeled using a binomial family parameter with a logit link, and count data used a Poisson family parameter with a log link. Regression coefficients and 95% confidence intervals (CI) were extracted using broom.mixed version 0.2.9.4 [24]. Missing data were addressed through complete case analysis, where case is defined as a visit within participant. Model diagnostics employed the modelDiagnostics function from multilevelTools version 0.0.1 [25], with model assumption violations addressed by comparing lmerTest results to Robust Scoring Equations estimator (RSE) results from robustlmm version 3.2.0 [26,27] using the Satterthwaite approximation from sjPlot version 2.8.14 [28]. RSE results were preferred when they differed substantially from the lmerTest analysis. The RSE method was not available for binary or count outcomes.

After data collection was complete, a post-hoc simulation-based power analysis [29] was conducted. This analysis used hypothetical population effect sizes and standard deviations [30] derived from the SBP and BMI models in Table 4 of Gabriel et al. [15] The goal was to estimate the sample size required to achieve 80% statistical power at an alpha level of 0.05 for parameter estimates assessing within-group changes from BL to 6wFU visits. For SBP, the power analysis indicated that 80% power could be attained with 32 participants. Adjusting the simulation to increase the magnitude of effect sizes associated with the visits by 10% resulted in achieving 80% power with a reduced sample size of 27 participants. For the exploratory secondary endpoint BMI, the analysis demonstrated that 80% statistical power could be reached with 27 participants.

The rates of change in clinical parameters were estimated using linear mixed-effect models [31] implemented with nlme version 3.1.160 [32]. Models included the clinical parameter as the dependent variable, day of study phase as a fixed effect, and participant ID as the random intercept. A random slope for day of study phase was included when supported by lower Akaike Information Criterion and Bayesian Information Criterion values. Separate models were fitted for prefeeding, fasting, and refeeding. For the body weight analysis, the fasting phase was divided into early (days 1–5) and late periods with phase, day of study phase, and their interaction included as fixed effects. Model assumptions were verified using residual and quantile-quantile plots. RSEs [26] with Satterthwaite approximation [28] were applied when necessary.

## 3. Results

### 3.1. Participant Characteristics

Baseline characteristics of participants (N = 29) are presented in Table 1. All participants had a diagnosis of stage 1 or 2 HTN, and 15 (52%) were taking anti-hypertensive medications. The most common pre-existing comorbidities were obesity (n = 15; 52%) and mixed and unspecified hyperlipidemia (n = 12; 41%) (Table 1 and Appendix A).

### 3.2. Treatment and Visit Characteristics

This study was conducted onsite at a residential fasting center, and participants’ adherence to the fasting and refeeding protocol was monitored by trained medical personnel as part of standard practice. All 29 participants completed BL, EOF, and EOR visits. Of the 29 participants, 26 (90%) completed at least seven consecutive days of fasting (see full description of treatment deviations in Section 3.4). The median (range) prefeeding, water-only fasting, and refeeding duration was 2 (1–4), 11 (7–40), and 5 (3–17) days, respectively.

Participants had daily access to nutrition and health education, were instructed to continue eating an SOS-free diet while offsite between EOR and 6wFU visits, and did not receive further dietary instruction thereafter. Of the 29 participants, 27 (93%) attended the 6wFU visit, which occurred at a median (range) of 45 (40–58) days after the EOR visit (Figure 1). Of the 27 participants, 17 (63%) attended the unanticipated 12mFU visit, which occurred at a median (range) of 364 (358, 383) days after the 6wFU visit. Of these 17 participants, 5 voluntarily visited the fasting center for additional interventions between 6wFU and 12mFU visits. Of these, 2 participants visited the fasting center one time and 3 participants visited two times, with a median (range) fasting and refeeding length of 10 (8, 13) and 6 (4, 9) days, respectively.

Participants’ adherence to an SOS-free diet during the follow-up periods was assessed using a previously described dietary screener with scores ranging from 0 (100% adherent) to 82 (0% adherent) [15]. Responses to individual questions are in Appendix A. The mean (SD) score was 12 (10), 6 (3), and 6 (4) at BL, 6wFU, and 12mFU, respectively. Participants reported increased fruit and vegetable consumption and decreased consumption of animal products, added salt, and added oil.

### 3.3. Adverse Events Identified During Prefeeding, Fasting, and Refeeding

Daily interviews and physical examinations, as well as weekly hematology, serology, urinalyses, and additional testing as indicated, were used to identify AEs from BL through EOR visits. AEs were classified into categories and grades using CTCAE v.5.0 [20]. We identified 453 AEs, of which 381 (84%) were mild Grade 1 (G1), 64 (14%) were moderate Grade 2 (G2), and 8 (2%) were severe Grade 3 (G3) events (Table 2). There were no Grade 4 (life-threatening), Grade 5 (death), or serious AEs. There were 11 (2%), 350 (77%), and 92 (20%) AEs during prefeeding, fasting, and refeeding, respectively (Table 2). G1, G2, and G3 were the highest-grade AE (HGAE) experienced by 7% (2/29), 69% (20/29), and 24% (7/29) of participants, respectively.

A detailed account of adverse event type, severity, prevalence, and outcomes can be found in Appendix A. The most commonly occurring AEs (i.e., experienced by >10% of participants) by treatment stage are shown in Appendix A. Overall, the five most common events were mild-to-moderate fatigue, mild decrease in blood bicarbonate, mild decrease in BUN/creatinine, mild-to-moderate high blood pressure, and mild-to-moderate nausea. During fasting, the three most common events were fatigue, hypertension, and decreased blood bicarbonate. There were 15 AE classifications that only occurred during fasting (Appendix A), including presyncope and various mild changes in blood chemistry and/or blood count identified by CMP and CBC analysis (Appendix A). During refeeding, fatigue was the most common event, and diarrhea occurred only during refeeding (Appendix A). As expected, four-fifths of hyperglycemia events occurred during refeeding. The other hyperglycemic event was identified at the EOF visit and may be due to an unreported protocol deviation.

Total AEs, associated system organ class, and event outcome are shown in Appendix A. The most frequently occurring G1 events were fatigue (n = 60/381), dizziness (n = 23/381), decreased blood bicarbonate (n = 22/381), decreased BUN/creatinine ratio (n = 18/381), nausea (n = 13/381), and decreased chloride (n = 13/381) and did not require intervention. Of the 381 G1 events, 33 were unresolved at EOR or 6wFU visits, and the outcome of 19/381 G1 events was unknown due to either unrepeated tests (9/19) or participant dropout (10/19). The most frequently occurring G2 events were fatigue (n = 19/64), hypertension (16/64), and presyncope (n = 10/64). All of the G2 events were resolved while onsite (n = 63/64) or before the 6wFU visit (n = 1/64). Of the eight G3 events, seven were hypertensive events, and one was a low neutrophil count. All G3 hypertensive events resolved before the EOR visit, but it is unknown if the low neutrophil count resolved due to participant dropout at the 6wFU visit.

We also assessed median changes in the clinical and laboratory markers used to assess AEs at each study visit. The median (IQR) for CBC variables remained within the normal range throughout this study (Appendix A), but several values significantly changed, and some changes were considered to be AEs. For example, there was a statistically significant increase in red blood cells at the EOF visit, and there were six G1 AEs due to increased red blood cells that did not require intervention (Appendix A). There were also several CMP values with statistically significant changes (e.g., decreased sodium at the EOF visit that resulted in 5 G1 hyponatremia events that did not require intervention, Appendix A). However, the only abnormal median (IQR) values included low BUN [6 (4, 8) mg/dL] for participants >60 years old at the EOR visit, low BUN/creatinine ratio at EOF [8 (7, 11) mg/dL] and EOR [7 (6, 9) mg/dL] visits for the entire population, and decreased carbon dioxide [17 (15, 20) mmol/L] at the EOF visit (Appendix A). Median values increased to the normal range for carbon dioxide at the EOR visit and for BUN and BUN/creatinine ratio at the 6wFU visit.

Urinalysis confirmed that during fasting, the concentration of ketones increased while pH decreased, and both values reverted to baseline upon refeeding when glucose metabolism resumed (Appendix A). At EOF, median urine volume (mL) decreased along with a slight increase in median (IQR) urine specific gravity (USG) to 1.008 (1.006, 1.010), which is well below the value (1.020) indicating dehydration (Appendix A). Although the standard USG data do not suggest that clinically significant dehydration occurred during fasting, dipstick analysis indicates that at EOF, four participants had USG > 1.020, suggesting mild dehydration (Appendix A). Median sodium (mmol/24 h) excretion also decreased to slightly below the reference range at EOF and EOR but normalized by the 6wFU (Appendix A). Median potassium (mmol/24 h) excretion was slightly elevated at baseline but decreased to low-normal at EOF, and was also normalized by the 6wFU. Pulse significant increase from BL to EOF visits (β (95% CI): 7.66 beats/min (3.10 to 12.21)), corresponding to an estimated rate of change of 0.55 beats/min per day, respectively, during fasting, but the changes were not clinically significant (Table 3, Table 4 and Appendix A). There were no clinically meaningful changes observed for BT and SpO2 (Table 3, Table 4 and Appendix A). These results indicate that fasting causes various temporary minor changes that are likely necessary to maintain homeostasis during the fasted state.

### 3.4. Deviations in Water-Only Fasting Intervention

As described in Section 3.2, 3/29 (10%) participants did not complete at least seven consecutive days of fasting. One fasted entirely on vegetable broth due to pre-existing cardiac arrhythmia. Of the other two participants, one temporarily interrupted the fast with a steamed potato/zucchini blend due to a G2 gastroesophageal reflux event, and one modified the fast with vegetable and fruit juice due to G2 hypoglycemia. Of the 26 participants, 17 completed at least seven consecutive days of fasting solely on water and 9 supplemented distilled water with vegetable broth (80 kcal/day) without interrupting ketosis. Six of those nine participants consumed vegetable broth while fasting because of a treatment-emergent AE, including G2 presyncope (n = 3), G1 gastroesophageal reflux (n = 1), G1 hypokalemia (n = 1), and G1 dry mouth (n = 1). The remaining 3 consumed vegetable broth for non-medical reasons. Additionally, of the 26 participants, 2 completed at least seven consecutive fasting days and had their fast prematurely terminated after day seven due to an AE, which included 1 participant with a G2 nausea event and 1 with concurrent G1 hypokalemia and G1 electrocardiogram QT-corrected prolonged events.

### 3.5. Changes in BW, Anti-Hypertensive Medication Use, and BP

BW and BP were also collected during daily vital signs monitoring throughout fasting and refeeding. At baseline (i.e., day 2 of daily rounding), 52%, 34%, and 14% had obesity, overweight, and normal weight, respectively (Table 1). At the EOR visit, 38%, 31%, 28%, and 3% had obesity, overweight, normal weight, and underweight, respectively (Appendix A). Baseline median (IQR) BW had a clinically meaningful reduction from 87.0 (80.0, 96.3) kg at BL to 79.1 (72.4, 87.8) kg and 81.4 (73.8, 88.7) kg at EOF and EOR visits, respectively (Table 3); after controlling for age and sex, the average changes from BL to EOF (β (95% CI): −7.76 kg (−8.85 to −6.66)) and from BL to EOR visits (β (95% CI): −6.59 kg (−7.69 to −5.49)) were statistically significant (Table 4). This corresponded to an average BW loss of −0.54 kg/day (*p* < 0.0001) while fasting, with higher rates of loss early in fasting and an average gain of 0.23 kg/day (*p* < 0.0001) while refeeding (Appendix A). The one person who was underweight at the EOF visit recovered to normal weight by the 6wFU (Appendix A). Median weight loss was also maintained at 6wFU and 12mFU visits (Appendix A).

The population recruited for this study had a current diagnosis of stage 1 or 2 hypertension. At the BL visit, 15/29 (52%) participants were taking nineteen anti-hypertensive medications; 11/15 were taking one, and 4/15 were taking two medications (Appendix A). Only 4/15 (27%) had achieved a medication-controlled SBP/DBP of <130/80 mmHg (Table 1). During the prefeeding period, all participants were tapered completely off anti-hypertensive medications by a trained physician according to their individual medical needs and remained off medications while at the residential fasting center. Participants’ medication use after leaving the fasting center was determined independently by their primary care physician. As such, no participants were taking any anti-hypertensive medications at EOF or EOR visits. Overall, baseline median (IQR) SBP/DBP had clinically meaningful changes from 135 (123, 150)/81 (7, 88) at BL to 125 (120, 131)/84 (80, 85) and 114 (108, 123)/78 (74, 82) mmHg at EOF and EOR visits, respectively (Table 3, Figure 2A,B). The estimated changes in SBP/DBP from BL to EOF and in SBP from BL to EOF were found to be statistically significant after controlling for age and sex (Table 4). The reductions in BP were sustained at the 6wFU and 12mFU visits. The estimated loss (95% CI) in SBP/DBP from the BL visit was −10.24/−4.65 (−16.04/−7.71, −4.46/−1.57) and −8.80/−8.83 (−15.47/−12.35, −2.15/−1.57) mmHg at 6wFU and 12mFU visits, respectively (Appendix A).Of the 15 medicated participants, 14 were taking anti-hypertensive medications at the BL visit and returned for the 6wFU visit; of them, only 1/14 was taking one anti-hypertensive medication at a reduced dose (Appendix A). Of the 17 participants, 7 attended the 12mFU visit and were taking anti-hypertensive medications; 5 were remedicated, and 2 were newly medicated (Appendix A). Figure 2C–F shows the SBP/DBP distribution of medicated and unmedicated participants at each study visit.

As previously described for a subset (n = 18) of this population [15], median values and estimated changes in these and other select cardiometabolic markers are also presented in Appendix A. At BL, only median total cholesterol, LDL, and FLI were outside of the normal limits. Median (IQR) total cholesterol normalized during refeeding to 4.71 (4.12, 5.52) mmol/L and was sustained at 6wFU and 12mFU visits. Median LDL remained slightly elevated throughout this study, and median triglycerides increased slightly out of normal limits at the EOR visit but normalized again at the 6wFU visit. Median insulin remained within normal limits throughout this study; however, the statistical model estimated a significant decrease from BL to EOF visit, a significant increase from BL to EOR visit and a return to BL by the 6wFU visit. Median (IQR) FLI was abnormally high at the BL visit (ST13) and mixed effects modeling provided evidence for a statistically significantly decrease from BL levels at 6wFU and 12mFU visits (ST14). These results suggest that the intervention effectively reduces high BP and BW, and may be an effective long-term management strategy for treating HTN, obesity, and other cardiometabolic disorders.

### 3.6. Water-Only Fasting and SOS-Free Diet Refeeding Treatment Acceptability

Participant acceptability of the entire fasting and refeeding intervention (BL to EOR) was assessed using the Treatment Acceptability/Adherence Screener (TAAS), with scores ranging from 10 (no acceptability/adherence) to 70 (full acceptability/adherence). Responses to individual TAAS questions are shown in Appendix A. Median (IQR) scores were 67 (63, 70) and 66 (60, 69) at EOF and 6wFU visits, respectively, suggesting a high degree of treatment acceptability.

Participant acceptability of foods permitted on an SOS-free diet was assessed using the Food Acceptability Questionnaire (FAQ), with scores ranging from 10 (no acceptability) to 70 (full acceptability). Responses to individual FAQ questions are shown in Appendix A. Median (IQR) scores were 47 (44, 52), 49 (45, 58), 48 (42, 55), and 45 (32, 70) at BL, EOR, 6wFU, and 12mFU visits, respectively. Together with the high study retention rate, these results suggest that participants found the medically supervised fasting and refeeding intervention to be highly acceptable.

## 4. Discussion

Here we present data that medically supervised, prolonged water-only fasting and whole-plant-food refeeding is well-tolerated, feasible, and potentially an effective treatment in people with multiple metabolic disorders. Importantly, the fasting and refeeding intervention was conducted at a residential fasting center using a protocol in which participants are thoroughly prescreened for potentially serious contraindications, have access to medical care 24 h per day, and are individually examined twice daily to identify potentially serious complications and, if necessary, modify the intervention accordingly. Using this protocol, more than 90% of participants in this study were able to complete at least seven consecutive days of fasting, and study attrition through the six-week follow-up visit was less than 10%. Participants also reported that the in-patient intervention was highly acceptable and preferable to conventional treatment with anti-hypertensive medications.

The most commonly identified AEs included transient mild (G1) or moderate (G2) fatigue, nausea, presyncope, dizziness, headache, and insomnia, which is in agreement with previously reported retrospective data from a larger mixed normotensive and hypertensive population using the same fasting protocol [14]. This study also utilized hematology and serology to thoroughly screen for AEs and identified some commonly occurring AEs not reported in our previous study. For example, 72% of participants experienced mild (G1) decreased blood bicarbonate while fasting, potentially due to the overproduction of ketoacids [33]. Symptoms of decreased blood bicarbonate include fatigue and nausea, which could potentially explain, at least in part, the high rates of fatigue and nausea commonly reported while fasting.

Seven of the eight severe (G3) events identified in this study were HTN and occurred during fasting, which is unsurprising given that this population had uncontrolled HTN and/or were weaned off anti-hypertensive medications before starting to fast. All of these HTN events were resolved while the participants were refeeding. Notably, fasting does not increase blood pressure in normotensive individuals [14,34]. The other G3 event was a low neutrophil count during refeeding that progressed from a G1 event during prefeeding. At the EOR visit, the participant’s WBC count had also decreased to a G1 event. Overall, we observed a trend of significantly decreased WBC count at EOR, but the median value did not decrease below the lower normal level, and abnormal values were not associated with any negative clinical outcomes. Previous research suggests that prolonged fasting induces stem cell regeneration and leads to a transient reduction in WBC count such that the body discards old WBCs before the immune system is replenished with new ones [35]. We do not know the relevance of this or other changes in CBC or CMP, but the results are in line with previous reports and warrant a more thorough analysis beyond the scope of this manuscript [36]. There were no other G3, life-threatening (G4), death (G5), or serious adverse events identified in this study. These data indicate that medically supervised fasting and refeeding are well-tolerated and low-risk in people with stage 1 and 2 HTN.

Similar to previous reports, we observed clinically meaningful and significant reductions in median SBP, DBP, BW, BMI, AC, and FLI [12,13,15,37,38] that correlated with the intervention and were sustained at the 6wFU [15]. Remarkably, median unmedicated blood pressure was normotensive by the EOR visit. At the 6wFU visit, only one participant had resumed anti-hypertensive medication use, and unmedicated median SBP/DBP was below 130/80 mmHg. Furthermore, the 17 participants who returned for the unanticipated 12mFU visit sustained a median BP below 130/80 mmHg and maintained median BW loss of >5%, suggesting the intervention is in alignment with clinical guidelines and may support long-term outcomes necessary to reduce CVD risk. The role of diet in maintaining outcomes after fasting remains elusive, but these participants were only moderately adherent to the recommended diet, suggesting that even with imperfect dietary adherence, outcomes achieved during fasting are durable. The long-term BW loss data are especially encouraging given the long-held belief that weight lost through fasting is quickly regained [39].

The mechanism by which fasting reduces blood pressure is not fully understood. One hypothesis is that dehydration during fasting results in hypovolemia, or reduced blood volume, that reduces blood pressure. Indeed, our data suggest that at least some participants were mildly dehydrated during fasting. Although this might explain the observed drop in blood pressure during fasting, it is noteworthy that these values began to revert to baseline during refeeding and had normalized by the 6wFU visit, whereas the reduction in blood pressure persisted.

This study has several limitations that should be addressed in future research. A major limitation is the potential for selection bias since the small number of participants were recruited from existing patients at a single fasting center and the majority of enrolled participants were post-menopausal women. As such, the results from this “self-selected” population may not be representative of the general hypertensive population. This study also lacks a diet control group and/or usual care comparison, and this study was not powered to assess changes in blood pressure or other cardiometabolic markers. Therefore, it is inconclusive what long-term results are attributable to diet alone and how the intervention compares to standard interventions (e.g., combination therapy anti-hypertensive medications). Furthermore, only 59% of participants re-enrolled for the 12-month follow-up visit. Future studies should aim to enroll and randomize a sufficient number of participants from a more general hypertensive population, include appropriate control populations, control for potential confounders such as prior diet and activity levels, and have additional planned study visits. Another important methodological limitation is that BP data are based on single rather than 24 h ambulatory BP measurements [40]. The effect of these methodological limitations on study outcomes is unknown, but these results are comparable to other reports in different populations [12,13,37].

## 5. Conclusions

Hypertension is the most common chronic disease worldwide, and when uncontrolled, it is a risk factor for developing cardiovascular and other diseases. Standard medical care typically includes anti-hypertensive medications that only minimally lower overall health risk, may be associated with serious side effects, have low adherence rates, and have a high lifetime cost [4,8,41,42]. Our results suggest that this in-patient, medically supervised, prolonged water-only fasting and SOS-free diet refeeding protocol may be a well-tolerated, low-risk alternative for long-term management of high blood pressure and obesity. These findings are in support of additional research to include a more diverse population, optimize treatment length and frequency, determine the effects of diet and other lifestyle modifications on sustained outcomes, and compare this intervention to standard treatment.

## Figures and Tables

**Figure 1 nutrients-16-03959-f001:**
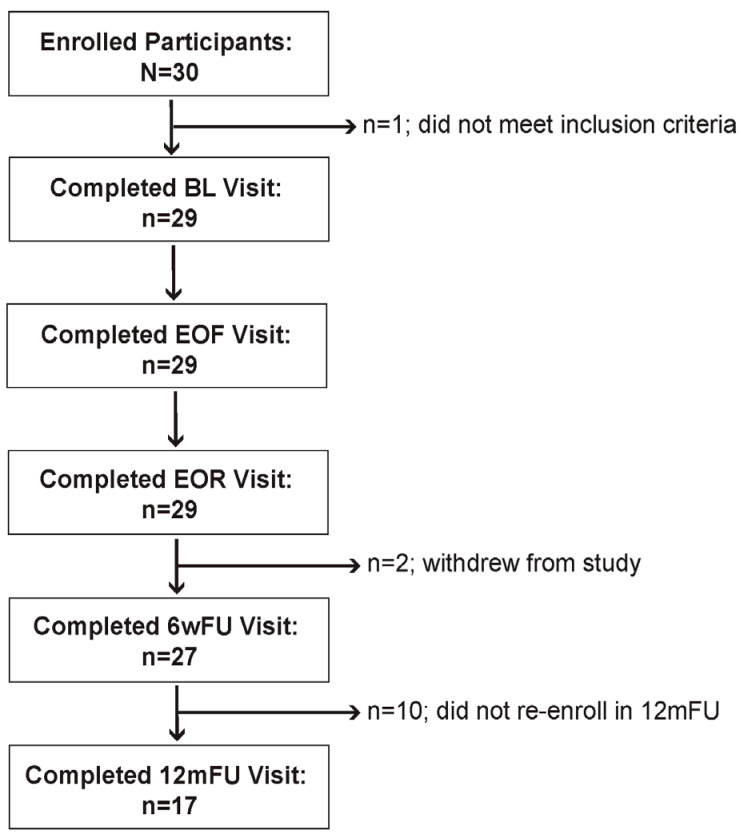
Enrollment and participation flow diagram. See “Participant Enrollment and Study Protocol” in the “Materials and Methods” section and “Participant Characteristics” in the “Results” section for details on eligibility and data collection. N, number of participants; BL, baseline; EOF, end-of-fast; EOR, end-of-refeed; 6wFU, six-week follow-up; 12mFU, 12-month follow-up. At EOF visit, 2/29 participants began refeeding before the blood draw and were excluded from serology and urinalysis. At the 6wFU visit, 2 out of 27 provided incomplete data: one provided all data except for anthropometric measurements, and one provided only blood pressure data.

**Figure 2 nutrients-16-03959-f002:**
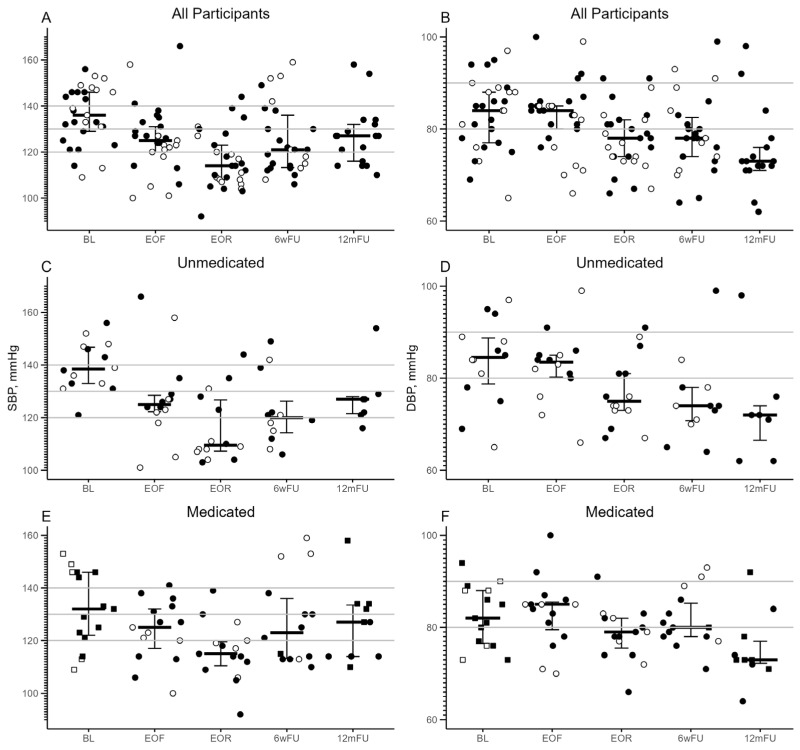
SBP (**A**,**C**,**E**) and DBP (**B**,**D**,**F**) by visit of all participants (**A**,**B**) and by baseline medication status (unmedicated (**C**,**D,**) and medicated (**E**,**F**)). Graphs include individual values as well as first (lower) and third (upper) quartiles and the median value. Circles and squares in panels (**C**–**F**) represent unmedicated and medicated SBP/DBP, respectively. Participants who attended the 6wFU visit (n = 27) and 12mFU visit (n = 17) are depicted as unfilled (open) and filled (black) symbols, respectively. Horizontal gray lines represent severity of HTN: SBP 120–129 mmHg, elevated blood pressure; SBP/DBP ≥ 130/80 mmHg, Stage 1 HTN; SBP/DBP ≥ 140/90 mmHg, Stage 2 HTN. A total of 3/29 and 12/29 participants did not provide data at 6wFU and 12mFU visits, respectively. SBP, systolic blood pressure; DBP, diastolic blood pressure; mmHg, millimeter mercury; BL, baseline; EOF, end-of-fast; EOR, end-of-refeed; 6wFU, six-week follow-up; 12mFU, 12-month follow-up; HTN, hypertension.

**Table 1 nutrients-16-03959-t001:** Select baseline characteristics.

Total Participants, N	29
Total Medicated ^^^ Participants, n (%)	15 (52)
Female, n (%)	19 (66)
Median (IQR) Age, y	62 (58, 67)
SBP < 130 AND DBP < 80 mmHg, n (%)	5 ^†^ (17)
SBP ≥ 130 AND ≤ 139 mmHg AND/OR DBP ≥ 80 AND ≤ 89 mmHg, n (%)	12 ^*^ (41)
SBP ≥ 140 AND/OR DBP ≥ 90 mmHg, n (%)	12 ^≠^ (41)
Normal Weight BMI, n (%)	4 (14)
Overweight BMI, n (%)	10 (34)
Obese BMI, n (%)	15 (52)

N, the total number of participants; n, a subset of 29 participants; %, percentage; IQR, interquartile range; y, years; SBP, systolic blood pressure; DBP, diastolic blood pressure: Stage 1 HTN = 130–139/80–89 mmHg, Stage 2 HTN = ≥140/90 mmHg; BMI, body mass index: normal weight, 18.5–<25 kg/m^2^; overweight, 25–<29 kg/m^2^; obese ≥30 kg/m^2^. ^^^ Medicated with anti-hypertensive medications; ^†^ 4 of 5 participants were taking anti-hypertensive medications. ^*^ 5 of 12 participants were taking anti-hypertensive medications. ^≠^ 6 of 12 participants were taking anti-hypertensive medications.

**Table 2 nutrients-16-03959-t002:** Total adverse events by grade and treatment phase.

	N (%)			
AE Grade	Prefeeding	Fasting	Refeeding	All Phases
1	9	289	83	381 (84)
2	2	54	8	64 (14)
3	0	7	1	8 (2)
4	0	0	0	0
5	0	0	0	0
Total	11 (2)	350 (77)	92 (20)	453 (100)

N (%), number and percentage of total adverse events; AE, adverse event; Grade 1, mild; Grade 2, moderate; Grade 3, severe; Grade 4, life-threatening; Grade 5, death. There were no Grade 4 or 5 events.

**Table 3 nutrients-16-03959-t003:** Vital signs by visit.

	Median (IQR)		
	BL	EOF	EOR
BW, kg	86.6 (80.1, 97.6)	79.1 (72.4, 87.8)	81.4 (73.8, 88.7)
SBP, mmHg <120	135 (123, 150)	125 (120, 131)	114 (108, 123)
DBP, mmHg <80	81 (76, 88)	84 (80, 85)	78 (74, 82)
BT, °C 36.1–37.2	36.7 (36.5, 36.9)	36.6 (36.5, 36.7)	36.7 (36.6, 36.9)
Pulse, min^−1^ 60–100	66 (60, 70)	73 (68, 83)	69 (64, 80)
SpO_2_, % 95–100	98 (97, 99)	98 (97, 99)	98 (98, 98)

Normal reference ranges are listed below each variable. At BL, EOF, and EOR, there were 29 participants. IQR, interquartile range; BL, baseline; EOF, end-of-fast; EOR, end-of-refeed; BW, body weight; DBP, diastolic blood pressure; mmHg, millimeter mercury; kg; BT, body temperature; °C, degrees Celsius; min^−1^, per minute; SpO_2_, saturation of peripheral oxygen; %, percent.

**Table 4 nutrients-16-03959-t004:** Significance of differences for vital signs.

	Estimates (95% CI) *p*-Value		
	EOF-BL	EOR-BL	EOR-EOF
Weight, kg ^‡^	−7.76 * (−8.85, −6.66) <0.001	−6.59 * (−7.69, −5.49) <0.001	1.17 * (0.07, 2.26) 0.038
SBP, mmHg	−9.34 * (−15.80, −2.89) 0.006	−18.93 * (−25.39, −12.48) <0.001	−9.59 * (−16.04, −3.13) 0.005
DBP, mmHg	0.71 (−2.76, 4.17) 0.693	−4.33 * (−7.79, −0.87) 0.017	−5.03 * (−8.50, −1.57) 0.006
BT, °C	−0.06 (−0.18, 0.06) 0.313	0.03 (−0.08, 0.15) 0.589	0.09 (−0.02, 0.21) 0.122
Pulse, min^−1^	7.66 * (3.10, 12.21) 0.002	5.00 * (0.45, 9.55) 0.036	−2.66 (−7.21, 1.90) 0.258
SpO_2_, % ^‡^	0.07 (−0.56, 0.71) 0.820	0.27 (−0.36, 0.89) 0.408	0.19 (−0.43, 0.82) 0.550

CI, confidence interval; BL, baseline; EOF, end-of-fast; EOR, end-of-refeed; BW, body weight; SBP, systolic blood pressure; DBP, diastolic blood pressure; mmHg, millimeter mercury; kg, kilogram; BT, body temperature; °C, degrees Celsius; min^−1^, per minute; SpO_2_, saturation of peripheral oxygen; %, percent. * Zero lies outside the 95% CI, so the finding is considered significant. ^‡^ Used robust mixed effects model on complete cases. Weight, SBP, and DBP models included 6wFU and 12mFU time points as described for Appendix A [15].

## Data Availability

The minimal dataset, excluding AE data, are publicly available at https://doi.org/10.5061/dryad.9cnp5hqt8. The AE data presented in this study are available on request from the corresponding author. The AE data are not publicly available due to privacy reasons.

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
