# Peer review of "Prolonged Water-Only Fasting Followed by a Whole-Plant-Food Diet Is a Potential Long-Term Management Strategy for Hypertension and Obesity"

_nutrients, 2024, doi:10.3390/nu16223959_

Round 1

Reviewer 1 Report

Comments and Suggestions for Authors

Ø  By using the turn-it-in via my institution I found 67% similarity. However, 62% was from a preprint of this work so I believe it is ok. 

Ø  Overall, the manuscript and each sub-section is good, well-written and well-presented.

A few recommendations:

·        Consider exploring whether different subgroups respond differently to the intervention. Discuss the role of other confounders such as prior diet and activity levels. Consider adjusting for these in the statistical models or mention how these factors were controlled for in the study design.

·        Distinguish statistical significance and clinical significance with further discussion. For example, explain whether the reduction in blood pressure and weight has meaningful clinical implications in the long term​.

·        Include a more detailed table summarizing the frequency, severity (e.g., mild, moderate, severe), and resolution of adverse events.

·        Discuss further the potential mechanisms underlying the effects of prolonged fasting on hypertension and obesity, referencing specific biomarkers or pathways that might be involved.

·        Discuss how the results may apply to broader populations!

·        Expand the limitations section by discussing potential biases such as self-selection and the lack of a control group.

·        Provide specific recommendations for future research.

Author Response

Thank you for your comments. Please find point-by-point responses below.

Comment 1: Consider exploring whether different subgroups respond differently to the intervention. Discuss the role of other confounders such as prior diet and activity levels. Consider adjusting for these in the statistical models or mention how these factors were controlled for in the study design.

Response 1: This study used a simple statistical approach, excluding subgroup analyses and interaction effects, due to the limited sample size. The power analysis indicated that this sample size was near the border of 80% statistical power for detecting main effects, making it inappropriate to pursue more complex analyses. An underpowered exploration of subgroup differences with the study’s sample size would lead to a higher risk for Type II errors, a lower ability to detect treatment effects, a reduced likelihood that statistically significant results reflect true effects, and an increased likelihood that the magnitude of the effect size is inflated.

To avoid the increased risk of overfitting that comes with a limited sample size, the study used minimal covariates. All models in Table 4 controlled for age and sex, as detailed in the methods section. For continuous outcomes (Table 4), models were tested with and without including baseline hypertension status as a fixed effect control variable. Since both approaches yielded similar results and met model assumptions, the more parsimonious model without hypertension status was reported. For binary outcomes, we retained baseline hypertension status as a covariate since it improved model diagnostics.

Although formal subgroup analyses were not feasible, we provided descriptive insights. Figure 2 shows how medication status and blood pressure changes across visits. Tables ST7 and ST12 show sex-stratified parameters. Table ST9 shows CMP parameters by sex and age group. Table ST14 shows study retention by weight class.

The limitations section acknowledges the study's limitations, specifically the lack of a control group. Future studies will address these limitations through inclusion of a control group and random assignment of participants to study arms. Additionally, future studies with larger sample size can investigate subgroup effects. The updated limitations section mentions that future studies should explore how controlling for potential confounders such as prior diet and activity levels influences treatment effects.

Comment 2: Distinguish statistical significance and clinical significance with further discussion. For example, explain whether the reduction in blood pressure and weight has meaningful clinical implications in the long term​.

Response 2: We agree that distinguishing between statistical and clinical significance of bp and weight outcomes is important and have attempted to meet this requirement in lines 525 to 533.     

Comment 3: Include a more detailed table summarizing the frequency, severity (e.g., mild, moderate, severe), and resolution of adverse events.

Response 3: These data are included in supplemental tables 3 and 4, which are overly large for the main text.

Comment 4: Discuss further the potential mechanisms underlying the effects of prolonged fasting on hypertension and obesity, referencing specific biomarkers or pathways that might be involved.

Response 4: We appreciate the interest in mechanism and, since the study is not address mechanisms, have included only a minimal discussion (lines 534 to 541).

Comment 5: Discuss how the results may apply to broader populations!

Response 5: This is addressed tangentially in the conclusion lines 560-567.

Comment 6: Expand the limitations section by discussing potential biases such as self-selection and the lack of a control group.

Response 6: This information is included in lines 542-558.

Comment 7: Provide specific recommendations for future research.

Response 7: This information is included in lines 567-570.

Reviewer 2 Report

Comments and Suggestions for Authors

Overview of the manuscript

The work focuses on the investigation of prolonged water-only fasting followed by a whole plant-food diet in patients affected by hypertension.  The authors apply a controlled protocol of patients care during the course of the clinical trial, in order to monitoring and adopt compensatory/recovering measures if adverse events occurred. Follow-up visits were organized at the 6-week and 12-month

Twenty-seven patients participate, hematochemical and urine parameters, blood pressure and body weight were monitored. The authors found that there was a sustained significant reduction of blood pressure, similarly a reduction of body weight, fatty liver index. The authors suggest that the proposed fasting/refeeding protocol is well tolerated and has low risks, therefore it can be

considered a valid option for lower and manage hypertension and related cardiometabolic disorders.

 GENERAL  COMMENT

The work is very interesting in proposing a new challenge in the management of hypertension and related cardiovascular disorders by opportune fasting and diet protocol. The work is well structured, and the application of the trial is described with great detail. The statistical protocol is adequate and adequately described in its complexity. The abstract should d be more concise and rephrased.

 Specific comments

Abstract

Pag. 1. Line 22-25: these details are redundant and not necessary in the abstract. Delete the paragraph.

Pag. 1. Line 26-28: The numerical details of your results are excessive in the abstract and make for not easy reading. Your data is adequately presented in the results section of your work.

Author Response

Thank you for your comments. Please find point-by-point responses below.

The abstract should d be more concise and rephrased.

Response 1: The abstract has been thoroughly revised based on your comments.

This single arm, pre-post interventional trial (clinicaltrials.gov, NCT04515095) investigates the safety, feasibility, and potential effectiveness of prolonged water-only fasting followed by a whole-plant-food diet in the long-term management of hypertension and other cardiometabolic disorders. Safety was assessed based on adverse events (AEs) that were recorded according to Common Terminology Criteria for Adverse Events (CTCAE) v5.0. Feasibility was assessed based on retention rate, ability to complete minimal fast length, and intervention acceptability. Twenty-nine participants with stage 1 and 2 hypertension and without type 2 diabetes were enrolled from a residential fasting center. Study retention was 100% at the end of refeed and 91% at the follow-up visit. Median (range) fasting and refeeding duration were 11 (7-40) and 5 (3-17) days, respectively, and >90% of participants were able to complete at least seven days of fasting. The majority of AEs were mild and transient and there were no higher-grade or serious AEs. At the end of the intervention, median systolic/diastolic blood pressure had normalized to below 130/80 mmHg, body weight reduced by >5%, and anti-hypertensive medication was completely discontinued. These results were sustained for at least six weeks and potentially up to one year. Our data suggest that the intervention may be a feasible, well-tolerated, low-risk option for lowering and managing high blood pressure, excess body weight, and other cardiometabolic disorders in people with stage 1 and 2 hypertension.

 Specific comments

Abstract

Pag. 1. Line 22-25: these details are redundant and not necessary in the abstract. Delete the paragraph.

Response 2: Agreed, lines have been deleted.

Pag. 1. Line 26-28: The numerical details of your results are excessive in the abstract and make for not easy reading. Your data is adequately presented in the results section of your work.

Response 3: Agreed those numerical details have been deleted.

Reviewer 3 Report

Comments and Suggestions for Authors

In this study authors assessd the tolerability and effectiveness of a food protocol consisting of  prolonged water-only fasting followed by whole plant-food refeeding,  in twenty-nine hypertensive subjects. They observed that at 6-week and 12-month follow-up, there were significant reductions in systolic and diastolic blood pressure, as well as improvements in other metabolic parameters.

The study is well conducted; the methodological approach is correct and described precisely.

The most important limitations are the absence of a control group and the small number of participants. However these two points are well underlined by the authors.

I have only minor observations:

Discussion

Regarding potential mechanisms by which the food protocol reduces blood pressure, authors state that ” one  hypothesis is that dehydration during fasting results in hypovolemia, or reduced blood volume, which may trigger the renin-angiotensin-aldosterone system (RAAS). This system increases sodium retention and decreases urine output, leading to a compensatory reduction in blood pressure. I strongly disagree with this hypothesis since it is well known that activation of the RAAS causes an increase rather than a reduction in blood pressure. Please review the text.

Table 3. Please provide p values (I suggest to add another column)   

Figure 2.  All the graphs that belong to this figure appear confusing and, in my opinion, are difficult to read. I suggest simplifying them, eliminating individual values

Author Response

Reviewer 3 Response 

Response: Thank you for your comments, please find point-by-point responses to your comments below.

I have only minor observations:

Discussion

Regarding potential mechanisms by which the food protocol reduces blood pressure, authors state that ” one  hypothesis is that dehydration during fasting results in hypovolemia, or reduced blood volume, which may trigger the renin-angiotensin-aldosterone system (RAAS). This system increases sodium retention and decreases urine output, leading to a compensatory reduction in blood pressure. I strongly disagree with this hypothesis since it is well known that activation of the RAAS causes an increase rather than a reduction in blood pressure. Please review the text.

Response 1: Thank you for pointing out this error, it had been removed from the text. 

Table 3. Please provide p values (I suggest to add another column)   

Response 2: Table 3 presents descriptive statistics and Table 4 shows results from the statistical models. Table 4 excludes p-values but includes 95% confidence intervals. The original draft of the statistical analysis section explained that 95% confidence intervals can be used to determine statistical significance:

Significance for continuous data was determined when CIs excluded zero, and for binary and count data, when exponentiated CIs excluded one.

Determining statistical significance in this manner corresponds to using a p-value of 0.05 as the threshold when testing for significance at the 0.05 alpha level. In Table 4, we presented confidence intervals rather than p-values since the confidence intervals can simultaneously communicate statistical significance and the level of uncertainty in the estimate.

We have included p-values along with CIs in the new Table 4 that corresponds with the median changes in Table 3.